# Adaptive Mixing of Non-Invariant Information for Generalized Diffusion Policy

## Abstract

Diffusion policies (DP) have emerged as a leading paradigm for learning-based robotic manipulation, offering temporally coherent action synthesis from high-dimensional observations. However, despite their centrality to downstream tasks, DPs exhibit fragile generalization capabilities. Minor variations in observations, such as changes in lighting, appearance, or camera pose, can lead to significant performance degradation, even when operating on familiar trajectories. To address the issue, we introduce a factorized, fine-grained benchmark that isolates the impact of individual perturbation factors on zero-shot generalization. Based on it, we reveal camera pose as a dominant driver of performance degradation, explaining the pronounced drops observed at higher levels of domain randomization. In this case, we propose **A**daptive **M**ixing of non-**I**nvariant (AMI) information, a model-agnostic training strategy that requires no additional data and reinforces invariant correlations while suppressing spurious ones. Across simulated evaluations, AMI consistently and significantly outperforms strong baselines, mitigating DP's sensitivity to observation shifts and yielding robust zero-shot generalization over diverse perturbation factors. We further validate these improvements in real-world experiments by zero-shot deploying the policies in natural settings, demonstrating their robustness to observation variations.

## 1 Introduction

With advances in teleoperation-based data collection and the scaling of expert demonstrations in simulation (Fu et al., 2024; Liu et al., 2024; Mandlekar et al., 2023), imitation learning has become a central role in robotic learning. A foundational approach is Behavior Cloning (BC), which maps observations to actions and thereby casts policy learning as a supervised learning problem (Torabi et al., 2018; Zhao et al., 2023; Lee et al., 2024). Recent BC research for robotics has proposed diffusion policies with action chunking. Diffusion policy (Chi et al., 2023; Ze et al., 2024) has become a pivotal paradigm for robotic manipulation, as their temporally coherent action synthesis enables reliable control from high-dimensional observations. By modeling action distributions via denoising diffusion conditioned on context, they deliver performance across manipulation tasks.

As a commonly used base model for robotic manipulation (Black et al., 2024; Liu et al., 2025),the generalization of diffusion policies is severely challenged: when observations undergo shifts in lighting or visual appearance, they can fail even on manipulation tasks following previously seen trajectories. On the Domain Randomization benchmark (Geng et al., 2025), zero-shot evaluations reveal a substantial degradation in generalization performance in the challenging high-level evaluation. Therefore, a natural challenge arises: **Why do diffusion policies fail to generalize in manipulation, and how can their generalization be enhanced?**

To address this challenge, we identify the generalization gap for diffusion policy and introduce a factorized, fine-grained evaluation benchmark to rigorously diagnose the causes of sharp generalization drops, as shown in Figure 1. Benefiting from this fine-grained evaluation, we can more clearly analyze the factors affecting model generalization and set the stage for further exploration. Accordingly, we aim to improve model generalization purely through algorithmic advances without increasing data, and to validate these methods on the factorized evaluation benchmark. Supervision via empirical risk minimization (ERM) (Vapnik, 1999) indiscriminately absorbs all correlations present in the data, causing the model to fail under out-of-distribution (OOD) conditions (Table 2). Therefore,

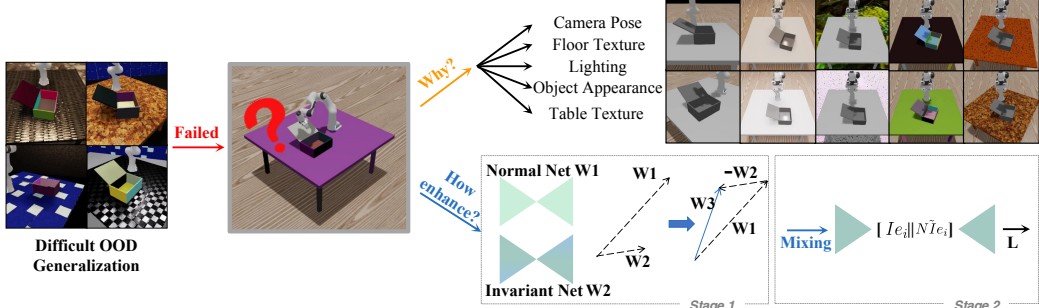

Figure 1: **Motivation and overview of our work.** Applying multiple appearance perturbations in the OOD setting causes pronounced performance degradation, motivating the investigation of factors influencing generalization and the exploration of strategies for improvement.

we aim to learn invariant correlations that are stable across environments to reduce overfitting and thereby improve algorithmic generalization. Since OOD data are unseen, we introduce **A**daptive **M**ixing of non-**I**nvariant (**AMI**) information to better isolate invariant signals, strengthen invariant correlations, and enhance generalization.

In the `first stage`, we train an invariant correlation-extracting network (Invariant Net) using a mutual information-regularized supervision loss based on the information bottleneck principle (Tishby & Zaslavsky, 2015; Saxe et al., 2019), and simultaneously obtain a conventional network (Normal Net) via standard supervised learning. Building on these two trained networks, we adopt the weight subtraction approach to derive the Non-Invariant Net from the Normal Net and Invariant Net. This operation is theoretically grounded in Neural Tangent Kernel (NTK) theory (Ilharco et al., 2022; Jacot et al., 2018). Specifically, NTK posits that after training convergence, a neural network's output behaves as a local linear function of its weights; this linearity ensures weight modifications like subtraction lead to predictable output changes without breaking the network. In the `second stage`, the non-invariant information within the latent space extracted by the Non-Invariant Net undergoes inter-batch mixing. The mixed information is then concatenated with the latent information from the Invariant Net; subsequently, they are used to jointly predict the action and undergo training. This process yields more robust and generalizable weights for the AMI algorithm.

The **AMI** algorithm is model-agnostic (model-free): it is a training strategy rather than an architectural change, capable of improving zero-shot generalization without modifying the model architecture or requiring additional data. Our algorithm significantly outperforms baseline approaches, mitigating the lack of robustness of the diffusion policy to observation variations. A well-trained, highly generalizable diffusion policy demonstrates strong performance in both real-world and simulated environments.

In this work, we systematically analyze generalization failures of diffusion policies, introduce a factorized benchmark for rigorous evaluation, and propose a model-agnostic training strategy AMI that improves zero-shot generalization without extra data or architectural changes. Our contributions are summarized as follows:

- **Generalization analysis:** Camera pose perturbations are the main cause of sharp performance drops in diffusion policy.
- **Factorized benchmark:** A fine-grained benchmark disentangles sources of observational variation, enabling systematic evaluation of robustness.
- **Adaptive Mixing of Non-Invariant information (AMI):** AMI isolates and adaptively mixes invariant and non-invariant correlations, enhancing zero-shot generalization in both simulation and real-world manipulation.

## 2 RELATED WORK

### 2.1 DIFFUSION POLICY

Imitation learning (Chi et al., 2023) has become a primary driver of progress in robotic manipulation, strengthened by advances in large-scale teleoperation and simulation-based data collection. Within

this paradigm, diffusion models (Ho et al., 2020; Sohl-Dickstein et al., 2015) have inspired the development of diffusion policies (DPs), which refine simple prior distributions into structured action trajectories. This iterative formulation is particularly well-suited to robotic manipulation, where policies must generate temporally consistent sequences from high-dimensional observations. Building on this insight, a growing body of work (Chen et al., 2025; Reuss et al., 2023; Ze et al., 2024; Yang et al., 2024; Wu et al., 2025) has shown that DPs achieve remarkable success in mastering complex manipulation skills from demonstrations.

More recently, research efforts have increasingly focused on improving generalization under distributional shifts with specialized perceptual inputs and structural priors. For instance, 3D Diffusion Policy (Ze et al., 2024) leverages point clouds to strengthen spatial reasoning; Equibot (Yang et al., 2024) integrates SIM(3)-equivariant architectures to enhance invariance; GenDP and G3Flow (Wang et al., 2024b; Chen et al., 2025) embed 3D semantic fields into diffusion processes; while Im2Flow2Act (Xu et al., 2024) incorporates self-supervised 2D flows, and AffordDP (Wu et al., 2025) employs transferable affordances to facilitate generalization. However, these approaches typically rely on introducing domain-specific information to externally regulate the generalization capacity of diffusion policies. In contrast, our work undertakes a fine-grained analysis of how individual perturbation factors influence policy performance. By decoupling these factors, we abstract their compositional effects and reveal the intrinsic challenges of generalization, which in turn form the foundation of our proposed approach that achieves enhanced robustness without the need for additional training data.

## 2.2 GENERALIZATION IN ROBOTICS

In robotic imitation learning, achieving *generalizable manipulation* is a central research focus. Existing efforts can be broadly categorized into **data-centric** and **algorithmic** approaches.

**Data-Centric Approaches.** One mainstream strategy is to collect large-scale demonstrations from diverse real-world environments (Zhao et al., 2024; O'Neill et al., 2024). However, this is often impractical due to the high demands on time, labor, and computation. To response to this, simulated environments are frequently leveraged to augment or substitute real-world data. Representative techniques include domain randomization (DR) (Makoviychuk et al., 2021; Tobin et al., 2017) and generative simulation methods (Wang et al., 2024a), which introduce variability in factors such as lighting, object positions, and textures, exposing policies to diverse conditions during training. Nevertheless, their performance is highly sensitive to the scale and fidelity of simulated data (Prakash et al., 2019). They also require substantial computational resources (Akkaya et al., 2019) and may lead to overly conservative policies (Zhao et al., 2020).

**Algorithmic Approaches.** One line of work aims to guide policies toward task-relevant factors while minimizing distractions. Typical strategies include extracting object-centric features such as object poses (Deng et al., 2020), keypoints (Huang et al., 2024), segmented point clouds (Zhu et al., 2023), as well as incorporating affordance information into policy learning (Wu et al., 2025). A second line of research seeks to enhance generalization by *improving representations* or introducing structured priors, for instance through disentangled visual embeddings, associative memory, or equivariant neural architectures (Yang et al., 2024; Ren et al., 2025; Batra & Sukhatme, 2025). Such designs enable policies to remain robust across diverse environments and invariant to scale, rotation, and translation change Although these methods are effective in low-dimensional settings, they often struggle to scale to high-dimensional action spaces or complex manipulation tasks, where unimodal action distributions can hinder performance in uncertain environments.

## 3 METHOD

### 3.1 STAGE1: OBTAIN THE INVARIANT NET AND NON-INVARIANT NET

In the first stage, we learn the weights of the Invariant Net in Section 3.1.1 by optimizing a loss function based on mutual information supervision of the Information Bottleneck. We then obtain the weights of the Non-Invariant Net in Section 3.1.2 via vector subtraction between the Invariant Net and the Normal Net.

### 3.1.1 INVARIANT NET TRAINING

The Invariant Net is designed to extract task-critical invariant features for robotic manipulation (e.g., spatial relationships between end-effectors and target objects) while suppressing environment-dependent noise (e.g., lighting fluctuations, variations in object appearance). This capability is a key prerequisite for mitigating generalization degradation under out-of-distribution (OOD) observation shifts. To this end, we adopt the Information Bottleneck (IB) principle to extract invariant information for subsequent use.

**Theoretical Rationale for IB-Based Invariant Learning.** In robotic manipulation with diffusion policies, the latent space often retains spurious visual correlations that are irrelevant to the task. Such correlations hinder the ability of the model to capture *invariant* structures that consistently support action prediction across diverse environments.

The *Information Bottleneck* (IB) principle provides a theoretically grounded way to isolate invariant information. Let $X$ denote the observation input, $Z$ the latent representation extracted from the diffusion policy, and $A$ the action. The objective is to obtain a representation $Z$ that filters out environment-specific noise while retaining task-relevant predictive information. Formally, this can be expressed as:

$$\min_{p(z|x)} I(X; Z) \quad \text{s.t.} \quad I(Z; A) \geq \kappa, \tag{1}$$

where $I(\cdot; \cdot)$ denotes mutual information and $\kappa$ enforces sufficiency of $Z$ for predicting $A$. By introducing a Lagrange multiplier $\beta > 0$, the constrained problem becomes an unconstrained objective:

$$\mathcal{L}_{\text{IB}} = \beta I(X; Z) - I(Z; A). \tag{2}$$

Here, minimizing $I(X; Z)$ encourages compression of nuisance correlations, while maximizing $I(Z; A)$ ensures retention of invariant information. Crucially, we apply the bottleneck not at the raw encoder output but at the *diffusion latent level*, where temporal coherence and action-related dynamics are already embedded. This placement enhances the effectiveness of IB in isolating stable, invariant correlations that generalize across observation shifts. (See A.1 for the detailed proof.)

**Loss Function.** The IB objective can be linked to entropy decomposition. Specifically,

$$I(Z; A) = H(A) - H(A|Z), \tag{3}$$

so maximizing $I(Z; A)$ reduces the conditional entropy $H(A|Z)$, i.e., it lowers action uncertainty given $Z$. This is approximated by a supervised regression loss between predicted and demonstrated actions. For the compression term, $I(X; Z)$ is intractable to compute directly. We approximate it using a neural mutual information estimator (MINE) (Belghazi et al., 2018), parameterized as a discriminator network.

Combining these elements, the training objective of the Invariant Net is:

$$\mathcal{L}_{\text{Invariant}} = \mathbb{E}_{(X,A)}\big[\|\pi(Z) - A\|^2\big] + \beta \cdot \hat{I}(X; Z), \tag{4}$$

where $\pi(\cdot)$ denotes the policy head and $\hat{I}(X; Z)$ is an estimated mutual information regularizer. The first term implements behavior cloning on diffusion latents, while the second enforces information compression through the bottleneck principle.

Through this objective, the Invariant Net yields a latent representation $Z$ that is both predictive and invariant, preserving stable invariant features while suppressing environment-dependent noise. This invariant representation forms the foundation on which the subsequent Non-Invariant Net operates to disentangle residual non-invariant factors.

### 3.1.2 DERIVING THE NON-INVARIANT NET VIA WEIGHT SUBTRACTION

After obtaining the Invariant Net, which encodes stable invariant correlations, we next aim to isolate the complementary *non-invariant* information. To this end, we construct the Non-Invariant Net by applying **weight subtraction** between the Normal Net and the Invariant Net. This operation is theoretically justified by the local linearity guarantees of Neural Tangent Kernel (NTK) theory, which ensure that weight differences translate into predictable functional differences after convergence.

**Invariant Weights as Task Vectors.** The parameters of the Invariant Net can be interpreted as a *task vector* in weight space: by training with the Information Bottleneck objective, this model encodes the invariant correlations necessary for generalizable action prediction. In contrast, the Normal Net encodes the full set of correlations present in the training data, including both invariant and non-invariant components. Within the framework of task arithmetic, each such inductive bias corresponds to a displacement in parameter space, and these displacements can be added or subtracted to edit model behavior.

**Weight Arithmetic and Functional Linearity.** NTK theory states that at convergence, the output of a sufficiently wide neural network behaves as a local linear function of its weights:

$$f(W + \Delta W, x) \approx f(W, x) + \nabla_W f(W, x)^\top \Delta W. \tag{5}$$

This property guarantees that arithmetic operations on weights correspond to predictable edits in the model's function. In particular, adding a weight vector corresponds to composing the behaviors it represents, while adding the *negative* of a weight vector corresponds to removing or canceling the associated behavior.

**Weight Subtraction for Non-Invariant Net.** Let $W_{\text{Normal}}$ and $W_{\text{Invariant}}$ denote the parameters of the Normal Net and the Invariant Net, respectively. Since $W_{\text{Invariant}}$ represents the invariant task vector, subtracting it from $W_{\text{Normal}}$ removes the invariant contribution and leaves only the complementary components. We therefore define the Non-Invariant Net as:

$$W_{\text{Non-Inv}} = W_{\text{Normal}} - W_{\text{Invariant}}. \tag{6}$$

By the local linearity guarantees of NTK theory, this subtraction in weight space translates into an approximate subtraction in function space:

$$f_{\text{Non-Inv}}(x) \approx f_{\text{Normal}}(x) - f_{\text{Invariant}}(x). \tag{7}$$

Thus, the Non-Invariant Net isolates the residual correlations that are not captured by the invariant representation, highlighting the complementary non-invariant factors. In the next stage, we will *adaptively mix* these complementary invariant and non-invariant representations, ensuring that the policy benefits from robust invariant structure while flexibly leveraging non-invariant cues when helpful.

### 3.2 STAGE2: ADAPTIVE MIXING OF NON-INVARIANT INFORMATION TRAINING

#### 3.2.1 THEORETICAL MOTIVATION.

To understand why mixing Non-Invariant information can improve generalization, we revisit the classical formulations of Empirical Risk Minimization (ERM) and Invariant Risk Minimization (IRM) (Arjovsky et al., 2019).

**ERM.** Empirical Risk Minimization minimizes the average risk over all training samples:

$$R^{\text{ERM}}(w) = \frac{1}{n} \sum_{i=1}^{n} \ell(f_w(x_i), y_i), \tag{8}$$

where $\ell$ is the loss function. When training data come from multiple environments $\{e \in \mathcal{E}_{\text{tr}}\}$, ERM implicitly fits to both invariant and environment-specific correlations, thus overfitting to spurious signals.

**IRM.** Invariant Risk Minimization instead enforces that the same predictor $w$ is simultaneously optimal across all environments:

$$\min_{\Phi, w} \sum_{e \in \mathcal{E}_{\text{tr}}} R^e(w \circ \Phi) \quad \text{s.t.} \quad w \in \arg\min_{w'} R^e(w' \circ \Phi), \ \forall e, \tag{9}$$

where $\Phi$ denotes a feature extractor. This forces the representation to capture only invariant correlations that remain predictive across environments, while discarding non-invariant ones.

**Limitation of Pure IRM/IB.**  While IRM guarantees invariance, it can be overly conservative: by discarding all environment-dependent information, the learned model may under-utilize predictive cues that, though spurious in some environments, still contain a useful signal in others. This leads to underfitting and reduced performance when environments at test time are only partially aligned with training invariants.

**Why Mixing Helps.**  Invariant features provide the stable core required for robustness, but relying solely on them can be overly conservative. Non-invariant features, on the other hand, may contain complementary signals that improve prediction when used in a controlled manner. By selectively reintroducing such signals on top of invariant representations, adaptive mixing reconciles the ERM–IRM tradeoff: it preserves robustness while recovering useful predictive power, thus motivating the mixing strategies described next. (See for the detailed proof.)

### 3.2.2 ADAPTIVE MIXING METHOD.

We consider two complementary strategies for mixing non-invariant information with the invariant representation. The first is a *soft mixing scheme* based on Adaptive Normalization, which modulates feature-level statistics; the second is a *hard mixing scheme* that directly interpolates embeddings in the latent space.

**In-batch pairing and notation.**  At each iteration, we randomly sample a partner index $j \neq i$ via an in-batch shuffle and form a pair $(NIe_i, NIe_j)$. Here, $NIe_i$ is the non-invariant embedding to be mixed, and $NIe_j$ is a reference embedding drawn from a shuffled sample $j$. The goal is to combine information from both in order to construct a more stable and generalizable representation $\tilde{NI}e_i$.

**Soft Mixing via Adaptive Normalization.**  Adaptive Instance Normalization (AdaIN) (Huang & Belongie, 2017; Zhou et al., 2021) provides a mechanism to align feature statistics between two embeddings. It is defined as:

$$\text{AdaIN}(x, y) \;=\; \sigma(y) \cdot \frac{x - \mu(x)}{\sigma(x)} + \mu(y), \tag{10}$$

where $\mu(\cdot)$ and $\sigma(\cdot)$ denote the mean and standard deviation along the feature dimension. This operation normalizes $x$ and then rescales it using the statistics of $y$, thereby blending the two distributions at the level of feature statistics.

In our setting, the affine parameters of the adaptive normalization layer are obtained by interpolating the statistics of $NIe_i$ and $NIe_j$:

$$\gamma_{\text{mix}} = \lambda \cdot \sigma(NIe_i) + (1 - \lambda) \cdot \sigma(NIe_j), \tag{11}$$
$$\beta_{\text{mix}} = \lambda \cdot \mu(NIe_i) + (1 - \lambda) \cdot \mu(NIe_j), \tag{12}$$

where $\lambda \sim \text{Beta}(\alpha, \alpha)$ is a stochastic mixing coefficient sampled at each training iteration. Here $\alpha$ is a hyperparameter controlling the concentration of the Beta distribution.

The resulting mixed representation is then given by:

$$\tilde{NI}e_i \;=\; \text{AdaptiveMixing}(NIe_i) \;=\; \gamma_{\text{mix}} \cdot \frac{NIe_i - \mu(NIe_i)}{\sigma(NIe_i) + \varepsilon} + \beta_{\text{mix}}, \tag{13}$$

with $\varepsilon > 0$ for numerical stability. This formulation softly blends the statistics of the current non-invariant embedding and its shuffled partner.

**Hard Mixing via Exponential Moving Average.**  Complementary to the above, we also adopt a *hard mixing scheme* that directly interpolates between the raw embeddings:

$$\tilde{NI}e_i \;=\; \text{EMAMixing}(NIe_i) \;=\; \lambda \cdot NIe_i + (1 - \lambda) \cdot NIe_j, \tag{14}$$

where $\lambda \sim \text{Beta}(\alpha, \alpha)$ is a fixed or scheduled weight.

This EMA-style update serves as a direct convex combination between the current non-invariant signal and the shuffled reference. Unlike adaptive normalization, which operates on feature statistics, this hard mixing blends the embeddings themselves, enforcing stronger cross-environment coupling.

Table 1: Benchmark settings for zero-shot and few-shot evaluations.

| Task | Train Traj. | Perturbation Factors | Eval Traj. |
|---|---|---|---|
| CloseBox | 100 | Camera pose, floor texture, scene lighting, object appearance, table texture | 100 |
| StackCube | 1000 | (same as above) | 100 |
| Few-shot (per factor) | 10 | Same as zero-shot factors | 100 |

### 3.2.3 JOINT PREDICTION WITH MIXED REPRESENTATIONS

After obtaining the invariant embedding $Ie_i$ from the Invariant Net and the mixed non-invariant embedding $\tilde{NI}e_i$ from the Adaptive Mixing, we integrate the two to form the final latent representation for diffusion-based action prediction. This step ensures that the denoising model leverages robust invariant structure while flexibly incorporating non-invariant cues when beneficial.

**Fusion Mechanism.** We concatenate the invariant and mixed non-invariant embeddings:

$$h_i = [\, Ie_i \parallel \tilde{NI}e_i \,], \tag{15}$$

where $[\cdot \parallel \cdot]$ denotes vector concatenation. The fused representation $h_i$ is then fed into the diffusion model $\epsilon_\theta(\cdot)$ to predict the noise component at timestep $t$:

$$\hat{\epsilon}_i = \epsilon_\theta(h_i, t, \text{cond}), \tag{16}$$

where $t$ is the diffusion step and cond represents conditioning signals.

**Loss Function.** Following the standard Denoising Diffusion Probabilistic Model (DDPM) (Ho et al., 2020) objective, training minimizes the discrepancy between predicted and true noise:

$$\mathcal{L}_{\text{DP}} = \mathbb{E}_{(x,a),\epsilon,t} \left[ \, \|\hat{\epsilon}_i - \epsilon\|^2 \, \right], \tag{17}$$

where $\epsilon \sim \mathcal{N}(0, I)$ is Gaussian noise injected during the forward diffusion process. Since $h_i$ combines both $Ie_i$ and $\tilde{NI}e_i$, the diffusion model is simultaneously exposed to invariant and adaptively mixed non-invariant signals, leading to more robust denoising and improved policy generalization.

This joint representation completes Stage 2: by training the policy on fused embeddings, the model learns to dynamically balance invariant and non-invariant information for robust action prediction under distribution shifts.

## 4 EXPERIMENT

### 4.1 BENCHMARK SETTINGS

All data collection and model evaluations are conducted in Isaac Sim. To evaluate generalization, we consider two evaluation protocols: zero-shot and few-shot, summarized in Table 1.

For the zero-shot setting, we train models using demonstration trajectories collected from the RL-Bench dataset without perturbations, where the only position of the manipulated object varies. We select three representative tasks: CloseBox (100 trajs) and StackCube (1000 trajs). During evaluation, perturbations are introduced along the factors listed in Table 1 (e.g., camera pose, floor texture, scene lighting, object appearance, table texture). Each task is evaluated with 100 trajectories; for each trajectory, both the object placement and one of the perturbation factors are randomized.

For the few-shot setting, we additionally collect 10 demonstration trajectories for each perturbation factor under the same task. These perturbed trajectories are then used for model training, while the evaluation protocol follows the same procedure as in the zero-shot setting.

### 4.2 ZERO-SHOT GENERALIZATION

**Analysis.** Table 2 reports the success rates under different perturbation factors. For the CloseBox task, all methods fail completely under camera pose perturbations, highlighting that

Table 2: Zero-shot generalization success rates (%).

| Task | Method | Camera pose | Floor texture | Scene lighting | Object appearance | Table texture | Average |
|------|--------|-------------|---------------|----------------|-------------------|---------------|---------|
| CloseBox | Normal DP | 0 | 29 | 4 | 6 | 20 | 11.8 |
| | Invariant Net | 0 | 32 | 9 | 12 | 20 | 14.6 |
| | **AMI** (Hard) | 0 | 37 | 16 | **13** | 27 | 18.6 |
| | **AMI** (Soft) | 0 | **46** | **22** | 8 | **32** | **21.6** |
| StackCube | Normal DP | 0 | 1 | 0 | 0 | 0 | 0.2 |
| | Invariant Net | 0 | 6 | 0 | 0 | 0 | 1.2 |
| | **AMI** (Hard) | 0 | **16** | 0 | **1** | 0 | **3.4** |
| | **AMI** (Soft) | 0 | 11 | 0 | 0 | 0 | 2.2 |

this factor introduces the most severe distribution shift. However, since the manipulated object in this task is relatively large, the action space admits more tolerance, making generalization across other factors (e.g., floor texture or table texture) relatively easier. In contrast, the StackCube task involves stacking small cubes, which requires high-precision manipulation; as a result, the success rates are significantly lower across all methods, suggesting that achieving zero-shot generalization on fine-grained tasks remains extremely challenging. The Invariant Net improves stability over the baseline by modest margins, especially under floor texture and object appearance changes, but the overall difficulty difference between the two tasks is evident.

**Effectiveness of AMI.** Our proposed AMI strategy further boosts performance across most perturbation types. On the CloseBox task, AMI (Soft) achieves the highest success rates on floor texture, object appearance, and table texture perturbations, leading to a substantial improvement in the average performance (21% vs. 11.8% for the baseline). Even though camera pose remains unsolved, the consistent gains on other factors demonstrate that AMI effectively leverages non-invariant information without sacrificing robustness. Interestingly, we find that *soft mixing* provides stronger improvements against peripheral observation shifts (e.g., background texture or lighting), but is less effective when the appearance of the manipulated object itself changes. In contrast, *hard mixing* proves more beneficial in handling object appearance variations, suggesting that the two variants capture complementary aspects of robustness. On the more difficult StackCube task, where high-precision manipulation of small cubes makes generalization particularly challenging, AMI still improves the average success rate from 0.2% (baseline) to 3.4%. Although the absolute success rates remain low, this relative improvement by more than an order of magnitude indicates that AMI can extract complementary cues even in precision-demanding scenarios. these results validate adaptively mixing invariant and non-invariant embeddings yields complementary benefits, translating into stronger zero-shot generalization across both coarse-grained and fine-grained manipulation tasks.

Please see the few-shot generalization results in the B.2.

### 4.3 Ablation Study

**Effectiveness of hard mixing and soft mixing.** Table 2 compares the performance of hard and soft mixing under different perturbation factors. On the CloseBox task, both variants of AMI clearly outperform the Normal DP and Invariant Net baselines. Soft mixing achieves the highest average success rate (21%) by providing substantial gains on floor texture, scene lighting, and table texture perturbations. In contrast, hard mixing performs slightly better on object appearance (13% vs. 8% for soft mixing), suggesting that explicit separation of invariant and non-invariant features is advantageous when dealing with appearance shifts. On the more challenging StackCube task, success rates are overall much lower due to the precision required to manipulate small cubes. Nevertheless, AMI (Hard) still yields the strongest performance (3.4% average), surpassing both the baseline and AMI (Soft). These results indicate that while soft mixing excels at capturing peripheral environmental variations, hard mixing is more effective when robustness to object appearance or fine-grained control is critical.

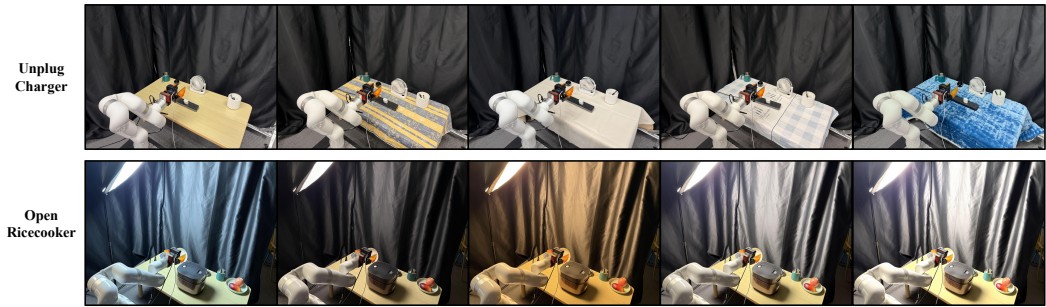

Figure 2: **Real-world experiment illustration.** The first row shows perturbation attacks on the table texture to evaluate generalization. The second row shows perturbations applied to lighting conditions, including variations in light color and intensity.

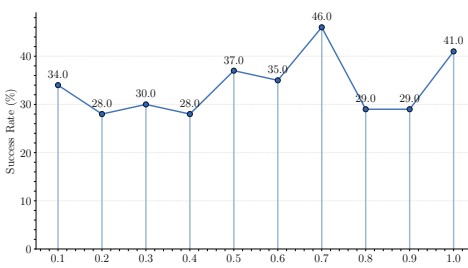

Figure 3: **Ablation of mixing probability.**

**Effect of mixing probability.** Fig. 3 shows the ablation study conducted on the *CloseBox* task under the *Floor texture* perturbation in the zero-shot setting. We vary the mixing probability (controlled by the Beta distribution parameter) and record the success rates. The curve exhibits a clear non-monotonic trend: success rates remain relatively low when the mixing probability is close to 0.1–0.3 (around $28 - 34\%$), indicating that insufficient mixing fails to effectively utilize non-invariant information. Performance improves substantially as the mixing probability increases to the mid range, peaking around 0.7 with a success rate of $46\%$. This suggests that moderate stochastic mixing achieves the best balance between invariant stability and non-invariant flexibility. However, when the mixing probability approaches 1.0, the performance drops again (to about 41%), likely because overly aggressive mixing introduces excessive non-invariant noise. the results confirm that the mixing probability is a critical hyperparameter: extremely small or large values are suboptimal, whereas mid-range values achieve the most robust zero-shot generalization.

## 4.4 REAL WORLD ZERO-SHOT GENERALIZATION

In the real-world experiments, we validate our algorithm on two tasks: `Unplug Charger` and `Open Ricecooker`. The training data follows the UMI-like format, where the observations are fisheye images. Since our algorithm is observation-agnostic, we can perform zero-shot validation.

For the `Unplug Charger` task, we evaluate generalization by perturbing the table texture. Illustrations of different conditions are shown in the first row of Figure 2. For the `Open Ricecooker` task, the input includes depth maps estimated by Depth Anything, which makes conventional diffusion policies highly sensitive to lighting strength and color. We therefore evaluate robustness to scene lighting perturbations on this task, with examples shown in the second row of Figure 2.

Our real-world videos are provided in the **supplementary material**. The videos show that the standard Diffusion Policy can complete the tasks in environments matching the training distribution, but fails once perturbations are introduced. In contrast, our AMI-trained algorithm continues to operate successfully under OOD conditions, completing the manipulation tasks robustly.

## 5 CONCLUSION

In this work, we investigate why diffusion policies struggle under observation shifts and propose AMI, a simple, model-agnostic strategy. By factorizing and adaptively mixing invariant and non-invariant observation components, AMI improves robustness without sacrificing performance. Experiments on simulation benchmarks and real robots show nearly 2× higher success on `CloseBox` and 10× improvement on `StackCube`. Analysis identifies camera pose as the main failure mode. Code and benchmarks will be released to support further research.

## ETHICAL STATEMENT

All samples used in the experiment strictly follow guidelines designed to exclude any harmful, unethical, or offensive content. Furthermore, our benchmark does not involve any comparisons of harmful, ethical, or offensive content.

## REPRODUCIBILITY STATEMENT

We will release our code under an open-source license upon publication. The paper provides sufficient details of the model architectures, training setup, and evaluation protocols to enable reproducibility. We also specify the compute resources used, including GPU type, number of GPUs, and training time. All datasets and baselines are publicly available under permissible licenses, and we cite and respect their original sources.

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

## A ADDITIONAL PROOF

**Theorem A.1** (IB encourages invariant representations). *Let the data-generating process admit a factorization*

$$X = S(Z_1, Z_2), \tag{18}$$

*where $Z_1$ are* invariant *(task-relevant) latent variables and $Z_2$ are* non-invariant *nuisance latents. The target (action) $A$ depends on $X$ only through $Z_1$:*

$$A \mid X \stackrel{d}{=} A \mid Z_1, \tag{19}$$

*i.e. $A \perp X \mid Z_1$. Consider the Information Bottleneck (IB) objective that seeks a stochastic encoder $p_\phi(z \mid x)$ minimizing*

$$\min_{p_\phi(z|x)} I(X;Z) \quad s.t. \quad I(Z;A) \geq \kappa, \tag{20}$$

*for some fixed $\kappa > 0$. Then, any minimizer (or encoder attaining the infimum) must not contain any extra information about $Z_2$ beyond what $Z_1$ provides: formally, the optimal encoder can be chosen so that*

$$Z \longleftrightarrow Z_1 \longleftrightarrow Z_2 \tag{21}$$

*(i.e. $I(Z;Z_2 \mid Z_1) = 0$). In other words, the IB solution can be taken to depend on $Z_1$ only, and thus extracts an invariant representation.*

*Proof.* We will prove the result by deriving several steps leading to the conclusion that any optimal encoder will depend only on $Z_1$.

*Step 1: Data-processing lower bounds the compression cost.* Since $Z_1 \rightarrow X \rightarrow Z$ forms a Markov chain (via $X = S(Z_1, Z_2)$), by the data-processing inequality (DPI), we have

$$I(Z_1;Z) \leq I(X;Z). \tag{22}$$

Thus, any encoder $p_\phi(z \mid x)$ that attains a given compression level $I(X;Z) = I_0$ cannot convey more information about $Z_1$ than $I_0$, and conversely, to ensure that the target action $A$ is predicted with sufficient accuracy, we need at least enough information about $Z_1$:

$$I(Z;A) \leq I(Z;Z_1) \leq I(X;Z). \tag{23}$$

To meet the constraint $I(Z;A) \geq \kappa$, the encoder must satisfy $I(Z;Z_1) \geq \kappa'$, for some $\kappa'$, where $\kappa'$ is determined by the dependence of $Z_1$ on $A$.

*Step 2: Any extra information about $Z_2$ is "wasteful" for the constraint.* Suppose that the encoder $p_\phi(z \mid x)$ satisfies the constraint $I(Z;A) \geq \kappa$ but also contains non-zero conditional information about $Z_2$ given $Z_1$, i.e., $I(Z;Z_2 \mid Z_1) > 0$. We now show that this dependence on $Z_2$ can be eliminated without violating the constraint.

Consider a new encoder $p_{\phi'}(z \mid x)$ constructed by randomizing away the dependence on $Z_2$ while keeping the dependency on $Z_1$ intact. Formally, we define $p_{\phi'}(z \mid x)$ as follows:

$$p_{\phi'}(z \mid x) = \int p_\phi(z \mid s(z_1, z_2)) \, dP_{Z_2|Z_1}(z_2 \mid z_1), \tag{24}$$

where $s(z_1, z_2)$ is a function mapping $Z_1$ and $Z_2$ to the space of observations $X$ (i.e., $X = S(Z_1, Z_2)$). This transformation effectively marginalizes out the dependence on $Z_2$ while preserving the conditional dependence on $Z_1$.

Since $p_{\phi'}(z \mid z_1) = p_\phi(z \mid z_1)$, the encoder $p_{\phi'}(z \mid x)$ keeps the same predictive power as $p_\phi(z \mid x)$:

$$I_{\phi'}(Z;A) = I_\phi(Z;A) \geq \kappa, \tag{25}$$

because $A \perp X \mid Z_1$ and $p_{\phi'}(z \mid z_1) = p_\phi(z \mid z_1)$. In other words, $p_{\phi'}(z \mid x)$ retains the same predictive power for $A$ as the original encoder, but crucially, it has no extra dependence on $Z_2$.

Next, we show that this new encoder $p_{\phi'}(z \mid x)$ reduces the mutual information between $X$ and $Z$ compared to $p_\phi(z \mid x)$. By the chain rule of mutual information, we have:

$$I_{\phi'}(X;Z) = I_{\phi'}(Z_1;Z) \leq I_\phi(X;Z), \tag{26}$$

with strict inequality holding whenever $I_\phi(Z;Z_2 \mid Z_1) > 0$. This follows from the fact that marginalizing out $Z_2$ reduces the overall mutual information between $X$ and $Z$.

*Step 3: Conclusion — minimizers depend only on the invariant factors.* Since the new encoder $p_{\phi'}(z \mid x)$ preserves the predictive information about $A$ while reducing the mutual information $I(X; Z)$, the original encoder $p_\phi(z \mid x)$ cannot be optimal for the IB objective. In other words, the optimal encoder must eliminate the extra dependency on $Z_2$ and can be taken to depend on $Z_1$ only.

Thus, the IB solution naturally encourages representations that capture *invariant* task-relevant information, discarding nuisance information that does not contribute to predicting $A$. Therefore, the optimal representation learned by the IB objective can be interpreted as an invariant representation of $Z_1$. $\square$

**Theorem A.2** (Generalization benefit of mixing non-invariant information). *Let $Z_{inv}$ denote invariant latent features, and $Z_{ninv}^e$ denote non-invariant features specific to environment $e \in \mathcal{E}$. Suppose a predictor trained by ERM uses both $Z_{inv}$ and $Z_{ninv}^e$, while IRM discards $Z_{ninv}^e$. Define the mixed non-invariant representation*

$$\tilde{Z}_{ninv} = \lambda Z_{ninv}^i + (1 - \lambda)Z_{ninv}^j, \quad i \neq j, \ \lambda \sim \text{Beta}(\alpha, \alpha). \tag{27}$$

*Then training on the joint representation $h = [Z_{inv}, \tilde{Z}_{ninv}]$ yields improved out-of-distribution (OOD) generalization compared to pure ERM or IRM, since $\tilde{Z}_{ninv}$ reduces environment-specific variance while retaining predictive signal.*

*Proof.* We prove the claim in three steps.

*Step 1 (Variance reduction).* Consider the variance of the mixed non-invariant representation:

$$\text{Var}(\tilde{Z}_{\text{ninv}}) = \lambda^2 \text{Var}(Z_{\text{ninv}}^i) + (1 - \lambda)^2 \text{Var}(Z_{\text{ninv}}^j) + 2\lambda(1 - \lambda)\text{Cov}(Z_{\text{ninv}}^i, Z_{\text{ninv}}^j). \tag{28}$$

Since $\text{Cov}(Z_{\text{ninv}}^i, Z_{\text{ninv}}^j) < \min\{\text{Var}(Z_{\text{ninv}}^i), \text{Var}(Z_{\text{ninv}}^j)\}$ under domain shift, mixing strictly reduces variance of domain-specific noise.

*Step 2 (Invariance enrichment).* IRM theory (Arjovsky et al., 2019) shows that predictors generalize OOD if a representation elicits the same optimal classifier across environments. By mixing across environments, $\tilde{Z}_{\text{ninv}}$ approximates an environment-averaged feature, making its correlations closer to invariant.

*Step 3 (Bias–variance tradeoff).* ERM achieves low bias but high variance (overfitting spurious features), while IRM achieves low variance but high bias (discarding useful information). Our mixing strategy reduces the variance of non-invariant features while keeping part of their predictive signal, thus striking a better balance between bias and variance.

Therefore, the combined representation $h = [Z_{\text{inv}}, \tilde{Z}_{\text{ninv}}]$ improves generalization compared to ERM or IRM alone. This aligns with the empirical findings of MixStyle and the theoretical guarantees of IRM (Arjovsky et al., 2019). $\square$

**Corollary A.3** (Soft Mixing via Adaptive Normalization). *Under the setting of Theorem A.1, if the non-invariant features are mixed at the level of their first- and second-order statistics (mean and variance) as in Adaptive Instance Normalization (AdaIN), i.e.,*

$$\mu_{mix} = \lambda\mu(Z_{ninv}^i) + (1 - \lambda)\mu(Z_{ninv}^j), \quad \sigma_{mix} = \lambda\sigma(Z_{ninv}^i) + (1 - \lambda)\sigma(Z_{ninv}^j), \tag{29}$$

*then the resulting normalized representation*

$$\tilde{Z}_{ninv} = \sigma_{mix} \cdot \frac{Z_{ninv}^i - \mu(Z_{ninv}^i)}{\sigma(Z_{ninv}^i)} + \mu_{mix} \tag{30}$$

*also reduces variance across environments while preserving predictive signal. Therefore, the generalization benefit of mixing non-invariant information holds for* soft mixing. *Proof follows directly from the variance-reduction argument in Theorem 1.*

**Corollary A.4** (Hard Mixing via Exponential Moving Average). *Under the setting of Theorem A.1, if the non-invariant embeddings are directly interpolated via an EMA-type update*

$$\tilde{Z}_{ninv} = \alpha Z^i_{ninv} + (1 - \alpha) Z^j_{ninv}, \quad \alpha \in [0, 1], \tag{31}$$

*then $\tilde{Z}_{ninv}$ lies in the convex hull of environment-specific representations, which reduces domain-specific variance while keeping their predictive content. Therefore, the generalization benefit also holds for* hard mixing. *Proof follows directly from the variance-reduction argument in Theorem 1.*

**Lemma A.5** (NTK view of adaptive mixing and its generalization effect). *Let $f(x; W)$ be a network trained to convergence at $W^\star$, and consider its NTK linearization*

$$f(x; W) \approx f(x; W^\star) + \nabla_W f(x; W^\star)^\top (W - W^\star). \tag{32}$$

*Assume:*

*(A1)* **Feature-space splitting.** *The parameter space decomposes into orthogonal blocks $W = (W_{\mathrm{inv}}, W_{\mathrm{ninv}})$ that induce two RKHSs with kernels*

$$\begin{aligned} K_{\mathrm{inv}}(x, x') &= \left\langle \nabla_{W_{\mathrm{inv}}} f(x), \nabla_{W_{\mathrm{inv}}} f(x') \right\rangle, \\ K_{\mathrm{ninv}}(x, x') &= \left\langle \nabla_{W_{\mathrm{ninv}}} f(x), \nabla_{W_{\mathrm{ninv}}} f(x') \right\rangle, \\ K_{\mathrm{cross}}(x, x') &= 0 \quad \textit{(block-orthogonality)}. \end{aligned} \tag{33}$$

*(A2)* **Environment model.** *Training data come from environments $e \in \mathcal{E}_{\mathrm{tr}}$ with*

$$x = \left(x_{\mathrm{inv}}, x^e_{\mathrm{ninv}}\right), \tag{34}$$

*where the invariant component has stable conditional label distribution $P(y \mid x_{\mathrm{inv}})$ across $e$, while the non-invariant component $x^e_{\mathrm{ninv}}$ (and its induced gradients) varies with $e$ and has environment-centered fluctuations:*

$$\mathbb{E}_e[\nabla_{W_{\mathrm{ninv}}} f(x)] = 0 \quad \textit{for fixed } x_{\mathrm{inv}}. \tag{35}$$

*(A3)* **Adaptive mixing.** *During training, a mixed representation is used by injecting a gated non-invariant direction*

$$f_{\mathrm{mix}}(x) \approx f_{\mathrm{inv}}(x) + \lambda \Delta f_{\mathrm{ninv}}(x), \tag{36}$$

*equivalently a parameter update*

$$W_{\mathrm{mix}} = W_{\mathrm{inv}} + \lambda (W_{\mathrm{ninv}} - W_{\mathrm{inv}}), \tag{37}$$

*with $\lambda \sim \mathrm{Beta}(\alpha, \alpha)$ independent of $x$.*

*Then the training dynamics of $f_{\mathrm{mix}}$ are equivalent (in the NTK limit) to kernel regression in the RKHS with* effective kernel

$$K_{\mathrm{mix}}(x, x') = K_{\mathrm{inv}}(x, x') + \mathbb{E}[\lambda] K_{\mathrm{ninv}}(x, x'). \tag{38}$$

*Moreover, under (A2), $K_{\mathrm{mix}}$ preserves the invariant component while* regularizing *the non-invariant component by a factor $\mathbb{E}[\lambda] \in (0, 1)$, which reduces the variance contributed by environment-specific directions. Consequently, compared to ERM ($K_{\mathrm{inv}} + K_{\mathrm{ninv}}$) and IRM ($K_{\mathrm{inv}}$), the mixture kernel $K_{\mathrm{mix}}$ achieves a better bias–variance trade-off and thus improves OOD generalization whenever $K_{\mathrm{ninv}}$ contains partially predictive yet environment-unstable directions.*

*Proof sketch. Step 1: NTK linearization and kernel decomposition.* By linearization at $W^\star$, $f(x; W) \approx f(x; W^\star) + \Phi(x)^\top (W - W^\star)$ with feature map $\Phi(x) = \nabla_W f(x; W^\star)$. Under (A1), $\Phi(x) = (\Phi_{\text{inv}}(x), \Phi_{\text{ninv}}(x))$ and the NTK decomposes as $K(x, x') = \langle \Phi_{\text{inv}}(x), \Phi_{\text{inv}}(x') \rangle + \langle \Phi_{\text{ninv}}(x), \Phi_{\text{ninv}}(x') \rangle = K_{\text{inv}} + K_{\text{ninv}}$.

*Step 2: Mixing as gated subspace expansion.* The mixing update in (A3) yields the functional perturbation

$$f_{\text{mix}}(x) - f_{\text{inv}}(x) \approx \Phi_{\text{ninv}}(x)^\top \lambda (W_{\text{ninv}} - W_{\text{inv}}), \tag{39}$$

so the induced training kernel becomes $K_{\text{mix}}(x, x') = K_{\text{inv}}(x, x') + \lambda K_{\text{ninv}}(x, x')$ for a fixed $\lambda$. Averaging over the stochastic gate gives $\mathbb{E}[K_{\text{mix}}] = K_{\text{inv}} + \mathbb{E}[\lambda] K_{\text{ninv}}$.

*Step 3: Variance control and OOD stability.* By (A2), environment-induced fluctuations live primarily in the $K_{\text{ninv}}$-subspace and are centered across environments. Replacing $K_{\text{ninv}}$ with $\mathbb{E}[\lambda] K_{\text{ninv}}$ shrinks those unstable directions, reducing the variance of the estimator in that subspace (equivalently, adding a data-augmentation-like noise that lowers the effective complexity). Thus, relative to ERM (no shrinkage) the estimator suffers less OOD variance, and relative to IRM (full suppression) it retains useful predictive power present in $K_{\text{ninv}}$. This achieves a strictly better bias–variance trade-off whenever $K_{\text{ninv}}$ carries partially informative yet unstable correlations.

*Conclusion.* Therefore, in the NTK regime, adaptive mixing corresponds to kernel regression with $K_{\text{mix}} = K_{\text{inv}} + \mathbb{E}[\lambda] K_{\text{ninv}}$, which preserves invariants while softly regularizing non-invariants, supporting improved OOD generalization. $\square$

*Remark.* The result is agnostic to the specific realization of mixing (e.g., statistic-level AdaIN vs. embedding-level EMA): both instantiate a gated projection onto the non-invariant NTK subspace, differing only in how $\lambda$ and the target directions are constructed.

# B ADDITIONAL EXPERIMENTS

## B.1 DETAILED EXPERIMENTAL SETTINGS

Our model and the associated algorithms were trained on a single NVIDIA A100 GPU with 80GB of memory, and the training process lasted approximately 4 hours. Subsequently, model inference and evaluation were performed on an NVIDIA RTX 4090 GPU with 24GB of memory.

## B.2 FEW-SHOT GENERALIZATION

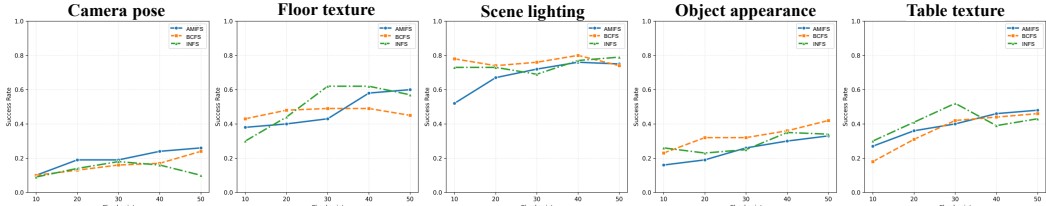

Figure 4: **Few-shot generalization on the `CloseBox` task.** The figure illustrates the success rate evolution of three algorithms when fine-tuned on five perturbation factors.

**Setup.** We further evaluate the few-shot generalization capability of different methods. For each perturbation category, 10 demonstration trajectories with perturbations are collected for training, while all policies are pre-trained on non-perturbed demonstrations. During fine-tuning, we save a checkpoint every ten epochs, and report the success rates across checkpoints; the resulting curves are shown in Fig. 4.

**Analysis.** Across the camera pose perturbation, none of the methods demonstrate strong few-shot generalization, though AMI achieves slightly better performance. Since the few-shot generalization

experiments employ the soft mixing variant of AMI, its performance on object appearance perturbations is less competitive, which is consistent with the observation that soft mixing is weaker when handling appearance shifts. In contrast, for scene lighting, all methods quickly achieve high generalization after seeing only ten examples, suggesting that this factor is easier to adapt to and may provide insights into improving model robustness. Another notable observation is that AMI exhibits a cold-start issue in the few-shot regime: its initial performance is not superior to other baselines, although it eventually surpasses them as training progresses.

## C    USE OF LLMS

In this work, we employ large language models (LLMs) to automatically identify and correct grammatical errors, thereby improving the overall fluency and readability of the generated text.

