# OpenReview forum: "Adaptive Mixing of Non-Invariant Information for Generalized Diffusion Policy"
_ICLR.cc/2026/Conference — Submitted to ICLR 2026_

### Official Review · Reviewer_JEan · 2025-10-24

**Soundness:** 1
**Presentation:** 2
**Contribution:** 2
**Rating:** 0
**Confidence:** 4

**Summary:**

The paper introduces a method for improving diffusion policies by increasing their generalization capabilities. This is achieved through different steps. First, the training of an invariant model which learns behavior based on mutual information loss. Second, the training of a normal diffusion model. Third, the training of a Non-Invariant model using both the invariant and normal model through inter-batch mixing. Two different approaches are evaluated for the mixing: Soft mixing via adaptive normalization and hard mixing via exponential moving averaging. The performance of the approach is evaluated zero- and few-shot on two tasks from the RL-Bench dataset and on two real robot tasks, where different pertubation factors are used.

**Strengths:**

**Strengths**

- Method is well explained and proven by mathematical formulations
- Problem and research gap are well motivated
- There are indications that the proposed method does improve performance for zero-shot generalization in specific pertubated enviroments

**Weaknesses:**

**Weaknesses**

- 4 pages only for the method is too long and restricts space that can be used for experiments, which results in a general problem of the paper structure
    - There are only 2 pages for experiment setup, experiments and analysis
    - Ablations should be moved to the appendix to make more room for further experiments
- Contributions are weak:
    - Generalization analysis: Camera pose perturbations are the main cause of sharp performance drops in diffusion policy.
        - There is no real analysis and this contribution is only based on two experiment setups in simulation
    - Factorized benchmark: A fine-grained benchmark disentangles sources of observational variation, enabling systematic evaluation of robustness.
        - The benchmark which was used is The Colosseum [1], as the pertubations introduced match exactly those introduced in this paper. This claim is therefore not valid. The paper itself is neither mentioned nor cited in the paper.
- Experiments and Ablations
    - The experiment section does not clarify which DP architecture is used for the evaluation
    - The whole method is only evaluated on two simulated and two real robot tasks
        - The real robot task results are not in the main paper and only available as video
        - It is also not clear how many rollouts were conducted on the real robot
    - The Ablation “Effect of mixing probability” is not clear in terms of results, as figure 3 shows decreasing and increasing values for high and low probabilities. It would be good to conduct more experiments and include a more in depth analysis why this behaviour occurs
    - Experiments using few-shot generalization indicate no superior performance of the proposed method
        - This raises the question if it is even worth to train two additional models, if a simple DP policy can be as good given a few training trajectories. This should be investigated and the overall usage of that method in few-shot generalization be better motivated.
    - As the method is independent of the actual model used, it would also be helpful to evaluate different DP architectures
    - Experiment evaluations should also be done on other benchmarks like RoboCasa, Calvin or Libero
- Minor weaknesses:
    - Figure 1 should be reworked to showcase how the proposed method works, instead of having it only as a small section of the figure
    - Section 2, 3 and 3.2 are empty sections, fill them with text or think about renaming/removing some subsections
    - 2.2 Data-Centric Approaches are missing actual Diffusion Policies which work with a lot of data to be more general
        - Especially VLAs using diffusion could be interesting in this regard
    - Table 1 can be removed and the contents be described in text
    - Line 365-366: RL Bench is not cited
    - Line 367: “three … tasks: …” this should be two
    - Line 371 sounds like new demonstrations were collected, but it should read that they were used from The Colosseum benchmark. Could you clarify this?
    - Table 2 add citation for normal DP


[1] Pumacay, Wilbert, et al. "The colosseum: A benchmark for evaluating generalization for robotic manipulation." arXiv preprint arXiv:2402.08191 (2024).

**Questions:**

This paper needs a major revision in terms of experiment evaluation and the concrete problem setting. The few-shot experiments are not convincing and distract from the zero-shot generalization story.

**Suggestions**
- Evaluate on more diverse tasks in The Colosseum setting
- Use different DP models
- Use further benchmarks
- Reduce the Method section and make it more comprehensible to have more space for experiments
- Move ablations into the appendix
- Include real-world robotics experiments as quantitative results in form of a table or figure in the main paper

---

### Official Review · Reviewer_x3wV · 2025-10-31

**Soundness:** 3
**Presentation:** 2
**Contribution:** 2
**Rating:** 2
**Confidence:** 3

**Summary:**

This paper introduces Adaptive Mixing of non-Invariant (AMI) information, a model-agnostic training strategy designed to enhance the generalization capabilities of Diffusion Policies (DP) for manipulation tasks. The authors first diagnose DP's fragility to observation shifts, identifying camera pose as a dominant factor in performance degradation, and propose a fine-grained evaluation benchmark to isolate these sources of variation. AMI has two stages: first, it uses the Information Bottleneck principle (measured with mutual informaiton between the observation, latetn,and action) to create networks that have invariant and non-invariant features for the task. Second, it adaptively mixes the non-invariant features using soft (Adaptive Normalization) or hard (EMA) mixing with the invariant features before final action prediction. Across sim and real experiments, the AMI approach outperforms the DP baseline, demonstrating robustness and generalization without requiring additional training data or architectural changes. Soft mixing enables robustness to peripheral shifts, and hard mixing enables robustness to visual changes in the object.

**Strengths:**

1. Categorising different variations (geometric and photometric) that DP fails at.
2. Addressing these by separating invariant and non-invariant features with separate networks. Using appropriate mixing techniques.

**Weaknesses:**

1. The authors show that geometric variations are the larger source of performance decrease in DP, but have chosen to solve the photometric variations; ignoring the bigger problem. That in itself is also OK, since it's possible that solving the geometric variation is an intractable problem today. However, the primary weakness I worry about is that they are not considering other methods that are specifically trying to solve this problem. In fact, (Batra & Sukhatme 2025) and (Yang et al., 2024), both mentioned in the related work, show examples of robustness to environmental variations such as the kind you are dealing with.
2. Authors mention that it is a model-agnostic training strategy. To substantiate this claim, It would help to show its performance on at least another model beyond DP.
3. Authors have only shown it on two sim tasks -- it would help for the supplementary material to have these rollouts, and to show it on more tasks. There are many task suites that can be used, such as RoboMimic and metaworld MT10.

**Questions:**

1. Please address comments in the weakness section.
2. You state that "we find that soft mixing provides stronger improvements against peripheral observation shifts (e.g., background texture or lighting), but is less effective when the appearance of the manipulated object itself changes. In contrast, hard mixing proves more beneficial in handling object appearance variations, suggesting that the two variants capture complementary aspects of robustness."
    But if an environment has both types of variations -- how do you propose to get robustness to both variations at the same time?

---

### Official Review · Reviewer_MoTy · 2025-11-01

**Soundness:** 1
**Presentation:** 2
**Contribution:** 2
**Rating:** 2
**Confidence:** 3

**Summary:**

This paper proposes AMI (Adaptive Mixing of Non-invariant Information), a model-agnostic approach to improve the zero-shot generalization of imitation learning policies under distribution shifts such as changes in lighting, appearance, or camera pose. AMI consists of two stages: (i) train a normal network and invariant network based on a mutual information objective, and subtract the invariant network from the normal network in weight space to obtain the non-invariant network; (ii) Combine empirical risk minimization (ERM) and invariant risk minimization (IRM) based on adaptively mixing non-variant and invariant features. The authors theoretically justify the weight subtraction in stage (i) based on neural tangent kernel (NTK) theory. They also propose two strategies for adaptive mixing: soft mixing using adaptive instance normalization and hard mixing using exponential moving average. Quantitative experiments are conducted in simulation, and the approach is demonstrated in a few real-world robot experiments.

**Strengths:**

- This paper studies a key problem concerning the robustness and generalization of imitation learning policies under minor distribution shifts.

- The proposed AMI method is well-explained in the paper and intuitive to follow, and the authors provide theoretical background for AMI.

**Weaknesses:**

- The real-world experiments are lacking quantitative results. The supplementary video only shows qualitative comparisons between DP and the proposed DP+AMI method. Furthermore, I have concerns about how the real-world experiment was conducted. Transitioning between time 2:27 and 2:32, the power connector appears to be closer to the robot when evaluating on DP+AMI while farther away for DP. However, the evaluation should only be considering “table texture change (OOD)” distribution shift.

- There is no description in the main paper or Appendix on hyperparameters or training details like network architecture. Was UNet or Transformer used for the DP backbone? What is the observation/action space? These are especially important, considering the paper claims a “factorized benchmark” as one of the main contributions of the paper in L97.

- There is no mention of prior work on benchmarking the robustness and zero-shot generalization of imitation learning policies under distribution shifts. L95 and L97 seem to claim this as two primary contributions of this paper. See Factor World [1], THE COLOSSEUM [2], or SimplerEnv [3] for a few notable examples of prior work in this direction.

- How does AMI compare to other algorithmic approaches such as those mentioned in Section 2.2? There are no baseline comparisons other than ablations on the proposed AMI method or the base DP method.

[1] https://arxiv.org/abs/2307.03659

[2] https://arxiv.org/abs/2402.08191

[3] https://arxiv.org/abs/2405.05941

**Questions:**

L367: There are only two tasks, not three?

Table 2: How many evaluation trials are conducted per shift?

Section 3.2.2: Would distributions other than the Beta distribution apply for adaptive mixing?

---

### Official Review · Reviewer_1yoF · 2025-11-06

**Soundness:** 3
**Presentation:** 2
**Contribution:** 3
**Rating:** 4
**Confidence:** 3

**Summary:**

This paper tackles the poor generalization of diffusion policies (DPs) in robotic manipulation, where small changes in lighting, textures, or especially camera pose cause large performance drops. The authors first introduce a factorized benchmark that isolates individual perturbation factors to diagnose what drives these failures. They find camera pose to be the dominant issue. To address this, they propose Adaptive Mixing of non-Invariant (AMI) information, a model-agnostic training strategy that improves robustness without needing extra data or architectural changes. AMI works by separating invariant and non-invariant representations, using the Information Bottleneck principle and Neural Tangent Kernel theory, and then adaptively mixing them to balance robustness and flexibility. Across simulation and real-world tests, AMI substantially improves zero-shot generalization (e.g., doubling success on “CloseBox” and increasing tenfold on “StackCube”), demonstrating that carefully mixing invariant and non-invariant cues can make diffusion policies far more robust to observation shifts.

**Strengths:**

1. The paper introduces a fine-grained, factorized benchmark that systematically isolates individual sources of distribution shift (e.g., lighting, texture, camera pose), offering clear insight into why diffusion policies fail.
2. The proposed AMI method improves zero-shot generalization without requiring architectural changes or additional data, making it broadly applicable across diffusion-based policies.
3. The method is supported by solid theoretical foundations, including the Information Bottleneck principle and Neural Tangent Kernel theory, which provide interpretability for how invariant and non-invariant features interact.
4. The paper demonstrates consistent and significant gains in both simulation and real-world robotic tasks, showing clear, quantitative improvements in robustness and transferability.

**Weaknesses:**

1. Although the paper includes real-world tests, the experiments are confined to only two simple tasks, leaving uncertainty about scalability to more complex or dynamic manipulation settings.
2. The approach involves multiple training stages and theoretical constructs (IB, NTK, mixing schemes) that may complicate implementation or reproducibility despite claiming model-agnostic simplicity.
3. The evaluation focuses mainly on standard diffusion policies and invariant variants, omitting direct comparisons with newer or stronger robustness-oriented diffusion or representation-learning methods.

**Questions:**

1. Your experiments primarily evaluate AMI on the CloseBox and StackCube tasks. How well do you expect the method to generalize to more complex, multi-object or contact-rich manipulation tasks where visual and spatial variations are more entangled?
2. Recent works such as [1] and [2] also explore generalization challenges in robotic imitation learning, including robustness to novel camera viewpoints and object arrangements. How does your approach relate to or differ from their analyses and strategies for improving generalization?
3. The ablation studies mainly vary mixing probability and compare soft vs. hard mixing. Could you provide more quantitative insight into performance variance, e.g., confidence intervals or statistical significance tests, to better assess the robustness and consistency of AMI’s gains across seeds or environments?

References:

[1] What Matters in Learning from Large-Scale Datasets for Robot Manipulation, Saxena et al., 2025

[2] Decomposing the Generalization Gap in Imitation Learning for Visual Robotic Manipulation, Xie et al., 2023

---

### Official Review · Reviewer_XFxb · 2025-11-07

**Soundness:** 1
**Presentation:** 2
**Contribution:** 2
**Rating:** 2
**Confidence:** 4

**Summary:**

The paper studies why diffusion policies struggle under observation shifts and proposes AMI, a model-agnostic training strategy to resolve this. AMI decomposes features into an Invariant Net learned with an information-bottleneck objective and a Non-Invariant Net obtained via weight subtraction, then mixes non-invariant information across batches before fusing with invariant features for action denoising. Using a factorized Isaac Sim benchmark that perturbs camera pose, textures, lighting, and object appearance, the author claims that AMI improves zero generalization on CloseBox and StackCube compared to a normally trained diffusion policy.

**Strengths:**

- The core idea of AMI, which uses an IB-regularized network and NTK-justified weight subtraction is novel.
- AMI is a flexible, model-agnostic recipe.
- The factorized benchmark isolates perturbation sources and identifies the most sensitive ones for diffusion policies, which makes the diagnosis interesting.

**Weaknesses:**

- The major concern of this paper is that the improvement over the naive diffusion policy seems limited. Specifically, in Table 2, 5/10 settings show no or minimal improvement (CloseBox-Camera pose, StackCube-Camera pose, Scene lighting, Object appearance, Table texture).

- While the paper identifies camera pose as the dominant failure mode, in the zero-shot experiments, the proposed AMI method also achieves 0% success on camera pose perturbations, showing no improvement over the baseline.

- The experiments were limited to two tasks in Isaac Gym, and no quantitative results are reported for the real-world tasks.

These make it difficult to be convinced that this method effectively improves the policy.

**Questions:**

- What is the architecture of the diffusion policy, and how is it trained? These details are missing in the paper.

- The paper claims that the method is model-agnostic. Have you tried using different model architectures to verify whether the results transfer as claimed?

---

### Meta-Review · Area_Chair_qtMo · 2026-01-05

**Summary:**

All author comments are below an accept, and were generally quite negative. Give the absence of any positive feedback for this paper, it is unlikely that the work would have been accepted. Some concerns included the limited improvement over the diffusion policy baseline, not justifying the complexity of the method, and non reproducibility to hyperparameters not being reported. Moreover, some claims (e.g. use of camera poses) seem to go unjustified.

**Reviewer Concerns:**

Negligible improvement over baseline Diffusion Policy; Limited Reproducibility; other concerns

**Reviewer Scores:**

All author comments are below an accept.

---

### Decision · Program_Chairs · 2026-01-26

Reject